# A Novel CaMKII Inhibitory Peptide Blocks Relapse to Morphine Seeking by Influencing Synaptic Plasticity in the Nucleus Accumbens Shell

**DOI:** 10.3390/brainsci12080985

**Published:** 2022-07-26

**Authors:** Zhuo Liu, Jianjun Zhang, Linqing Miao, Qingyao Kong, Xiaodong Liu, Longchuan Yu

**Affiliations:** 1People’s Public Security University of China, Beijing 100038, China; 20052372@ppsuc.edu.cn; 2CAS Key Laboratory of Mental Health, Institute of Psychology, Beijing 100101, China; zhangjj@psych.ac.cn; 3Beijing Advanced Innovation Center for Intelligent Robots and Systems, Beijing Institute of Technology, Beijing 100081, China; linqingmiao@bit.edu.cn; 4Department of Anesthesia and Critical Care, The University of Chicago, Chicago, IL 60637, USA; qingyao@uchicago.edu; 5Beijing Institute of Chinese Medicine, Beijing University of Chinese Medicine, Beijing 100029, China; 6School of Life Sciences, Peking University, Beijing 100871, China; yulc@pku.edu.cn

**Keywords:** long-term depression, Ca^2+^/calmodulin-dependent protein kinase II, relapse, reinstatement, nucleus accumbens shell, morphine self-administration, synaptic plasticity, opioids

## Abstract

Drugs of abuse cause enduring functional disorders in the brain reward circuits, leading to cravings and compulsive behavior. Although people may rehabilitate by detoxification, there is a high risk of relapse. Therefore, it is crucial to illuminate the mechanisms of relapse and explore the therapeutic strategies for prevention. In this research, by using an animal model of morphine self-administration in rats and a whole-cell patch–clamp in brain slices, we found changes in synaptic plasticity in the nucleus accumbens (NAc) shell were involved in the relapse to morphine-seeking behavior. Compared to the controls, the amplitude of long-term depression (LTD) induced in the medium spiny neurons increased after morphine self-administration was established, recovered after the behavior was extinguished, and increased again during the relapse induced by morphine priming. Intravenous injection of MA, a new peptide obtained by modifying Ca^2+^/calmodulin-dependent protein kinase II (CaMKII) inhibitor “myr-AIP”, decreased CaMKII activity in the NAc shell and blocked the reinstatement of morphine-seeking behavior without influence on the locomotor activity. Moreover, LTD was absent in the NAc shell of the MA-pretreated rats, whereas it was robust in the saline controls in which morphine-seeking behavior was reinstated. These results indicate that CaMKII regulates morphine-seeking behavior through its involvement in the change of synaptic plasticity in the NAc shell during the relapse, and MA may be of great value in the clinical treatment of relapse to opioid seeking.

## 1. Introduction

Drugs of abuse induce pathological changes in the brain reward circuits, especially the mesolimbic dopamine system, leading to mental and behavioral disorders, including drug craving, compulsive drug use, and relapse [1]. According to DSM-V (the fifth edition of the Diagnostic and Statistical Manual of Mental Disorders), Opioid Use Disorder is characterized by compulsive use of opioids, craving, tolerance, withdrawal syndrome, repeated relapse, and other features [2]. One of the hallmarks of Opioid Use Disorder is the long-term likelihood of relapse, despite treatment [3].

A large number of studies have suggested that drug use-evoked changes in synaptic plasticity of the brain reward circuits, which outlast the drug present in the brain and contribute to the neural circuits’ reorganization [4], are important mechanisms of drug addiction and relapse [2,5,6,7]. Repeated drug use alters the synaptic mechanisms that normally serve reward-related learning and memory [8,9,10,11,12,13,14] and forms abnormal drug-related memories, which lead to a high risk of relapse [8,15,16,17,18,19].

The molecular and cellular mechanisms underlying synaptic plasticity are found to be involved in drug addiction and play key roles in the relapse [13,19,20,21,22,23,24,25,26,27,28,29,30]. For example, drug usage altered two well-known important mechanisms of synaptic plasticity, long-term potentiation (LTP), and long-term depression (LTD), in many brain regions [9,11,31,32,33,34,35,36,37,38,39,40,41]. Meanwhile, regulating synaptic plasticity influenced drug-seeking behavior [19,26,28,42]. Therefore, to develop good therapeutic strategies to prevent relapse to drugs of abuse, it is essential to illustrate how the synaptic plasticity is altered during drug addiction and the molecular mechanisms involved in this course.

Many molecules in the brain are involved in the mechanisms of synaptic plasticity, one of which is Ca^2+^/calmodulin-dependent protein kinase II (CaMKII). CaMKII is crucial in the processing of learning and memory [43,44,45,46] as well as the mechanisms of LTP and LTD [47,48,49,50,51,52,53,54]. In the brain, the α- and β-subunits of CaMKII are the predominant isoforms and form mostly dodecameric holoenzymes that are composed of either one or both subunit types [55]. Weak stimuli activate the kinase through the binding of Ca^2+^/calmodulin without autophosphorylation. If the duration or magnitude of the Ca^2+^ elevation is greater, two or more subunits are activated on the same holoenzyme. In this case, an activated CaMKII subunit is phosphorylated on Thr286 (this numbering refers to the α-isoform and is Thr287 in the β-isoform) by its neighboring active subunit. Once phosphorylated, CaMKII has Ca^2+^-independent “autonomous” activity [56]. When a threshold number of kinases are phosphorylated, the rate of autophosphorylation exceeds the rate of dephosphorylation, leading to the long-term persistent activation of CaMKII [55]. In other words, an increase in the phosphorylation of CaMKII means an increase in the kinase activity. Moreover, other mechanisms make CaMKII generate Ca^2+^-independent activity, in which binding to the NMDA-receptor subunit GluN2B mediates the accumulation of CaMKII at excitatory synapses in response to LTP stimuli [56]. Different degrees of activation and persistent activity of CaMKII provide a molecular basis for processing synaptic memory, indicating its essential role in the transition to drug addiction and relapse.

The NAc is the main component of the reward circuits and plays a critical role in reward-related behaviors [57] and substance addiction, including alcohol [58]. In the NAc, over 95% of the neurons are GABAergic Medium Spiny Neurons (MSNs), which receive dopaminergic afferents mainly from the Ventral Tegmental Area and convergent glutamatergic inputs from several subcortical brain regions and prefrontal cortex [2]. Based on the anatomical composition, the NAc can be further divided into two distinct subregions: the core and the shell, which receive differential inputs and participate in various processes related to drug addiction [59]. Therefore, it is suggested that CaMKII activity in the NAc is essential to addiction behavior and relapse.

Many studies have demonstrated that CaMKII in the NAc shell is involved in drug addiction. For example, injection of CaMKII inhibitor KN-93 into the NAc shell attenuates cocaine-seeking behavior [60], while chronic cocaine treatment activates the transcription of αCaMKII in the NAc shell, enhancing the reinforcement of cocaine [61]. Moreover, αCaMKII overexpression in the NAc shell increases animals’ response to amphetamine [62], and decreases of phosphorylated αCaMKII at Thr286 by transient expression of a dominant-negative αCaMKII mutant K42M in the NAc shell persistently blocked the enhanced locomotor response to amphetamine and self-administration [63]. Similarly, cue-induced reinstatement of alcohol seeking is associated with increased CaMKII phosphorylation on Thr286 in the NAc shell of mice [64]. Consistent with the above findings, our previous studies also demonstrated that αCaMKII phosphorylated on Thr286 in the NAc shell increased during priming-induced morphine-seeking behavior in rats [30], and inhibiting CaMKII activity in the NAc shell with “myr-AIP” blocked the reinstatement to morphine seeking [29]. However, it is still unclear how CaMKII in the NAc shell regulates morphine-seeking behavior during the reinstatement.

We hypothesize that drug-evoked rapid changes of synaptic plasticity mediated by CaMKII in the NAc shell may underlie the relapse to opioids even after long periods of withdrawal. Therefore, in this paper, we studied the changes in synaptic plasticity at the multiple stages of morphine addiction, especially the relapse, and explored whether and how CaMKII was involved in the course.

## 2. Materials and Methods

### 2.1. Subjects

Male Sprague–Dawley rats, which weighed 220–250 g on arrival, were purchased from the Laboratory Animal Center of the Academy of Military Medical Sciences in Beijing (China). Rats were housed in a reverse-cycle room (12 h light/dark) with food and water available *ad libitum* except for the training days of morphine self-administration. All rats were afforded at least seven days of acclimation before the start of experimental procedures.

### 2.2. Surgery

As previously reported [29,30], before surgery, rats were anesthetized with 75 mg kg^−1^ of sodium pentobarbital intraperitoneally. An indwelling silastic catheter (AniLab, Ningbo, China) connected to a polyethylene (PE) catheter (PE-50, Instech Solomon, Plymouth Meeting, PA, USA), was inserted into the right jugular vein and sutured in place. The PE catheter was then threaded subcutaneously over the scapula. We flushed the catheter daily with 0.4 mL heparinized saline (100 IU mL^−1^). During the first four days following surgery, rats received 160,000 IU of penicillin. After surgery, rats had at least seven days for recovery.

All animal procedures were performed in accordance with the guidelines provided by the Regulation for the Administration of Affairs Concerning Experimental Animals (China, 1988) and approved by the Research Ethics Review Board of the Institute of Psychology, Chinese Academy of Sciences (A15016).

### 2.3. Drugs

Morphine hydrochloride injection (Shenyang First Pharmaceutical Factory, Shenyang, China) was diluted in sterile saline. Our previous studies have demonstrated that inhibiting CaMKII activity in the nucleus accumbens shell with inhibitor “myr-AIP” blocked the reinstatement to morphine-seeking behavior [29], which indicates CaMKII inhibitor may be of great value to preventing relapse to opioids clinically. In the present study, we modified myr-AIP with poly-arginine [65] and got a novel peptide named “MA”, which can be administrated intravenously. MA (myr-βAla-Arg-Arg-Arg-Arg-Arg-Arg-Arg-Lys-Lys-Ala-Leu-Arg-Arg-Gln-Glu-Ala-Val-Asp-Ala-Leu) was synthesized by GL Biochem (Shanghai, China).

### 2.4. Morphine Self-Administration (SA)

#### 2.4.1. Apparatus

As previously reported [29,30], sixteen 29 × 29 × 26 cm operant chambers (AniLab, Ningbo, China) were used. Each chamber had a white house light for illumination. Two holes located in a sidewall were each equipped with a blue LED. A speaker outside the chamber provided auditory stimuli. A liquid dipper with a recessed magazine, which is located between the two holes, was used in tests for a natural reward. The drug solution was delivered through Tygon tubing, which was encased in a protective metal spring leash and connected to a pump-driven syringe. Numbers of “nose-pokes” and drug infusions were recorded by a computer via an interface (AniLab, Ningbo, China).

#### 2.4.2. Training

As previously reported [29,30], rats were trained to poke a hole for intravenous morphine injections on a fixed-ratio (FR) 1 schedule. Each day included one training session lasting 3 h. When the session began, the house light was on. Poking the active hole led to a morphine infusion (0.3 mg kg^−1^) accompanied by the light-tone mixed stimulus as the cue. Poking the inactive hole had no scheduled consequences. A 15-s time-out followed morphine infusion. The chamber was completely dark during the time-out, and poking any hole had no consequences. The house light was turned on again after the time-out. Each rat was allowed 20 g of rat chow during training days.

#### 2.4.3. Extinction

Each day included one extinction session lasting 3 h. The extinction procedure was the same as the training session, except that poking the active hole did not lead to any morphine injection.

#### 2.4.4. Reinstatement

Immediately after receiving an intraperitoneal injection of 5 mg kg^−1^ morphine, rats were placed in the operant chamber. The procedure of reinstatement was the same as the extinction session. A reinstatement session lasted 3 h.

### 2.5. Saccharin SA

#### 2.5.1. Training and Extinction

As previously reported [30], rats were trained to poke the nose into a hole for 0.2% (wt/vol) saccharin solution under an FR1 schedule during their dark cycle. The training and extinction procedures were the same as the morphine SA, except that poking the active hole led to 0.1 mL saccharin delivery instead of morphine infusion. Each training session and extinction session lasted only 1 h.

#### 2.5.2. Reinstatement

Saccharin seeking was reinstated with two 0.1 mL saccharin deliveries. The session began with a noncontingent saccharin delivery. The interval between the two deliveries was 10 min. The reinstatement procedure was the same as for extinction except for the two saccharin deliveries. The session lasted 1 h.

### 2.6. Locomotor Activity

Operant chambers were equipped with photocells for quantifying locomotor activity. The photocells were spaced evenly along the longitudinal axis of the chamber and positioned 2.5 cm above the floor. Separate interruptions of photocell beams caused by the movement of rats were detected and recorded. The total number of beam breaks represents the level of locomotor activity.

### 2.7. Behavior Experiments

#### 2.7.1. Acquisition, Extinction, and Reinstatement of Morphine Self-Administration

Acquisition: Rats were trained to self-administer morphine over 18 consecutive training sessions (one per day). During the phase, rats learned to acquire morphine by themselves, and the number of active pokes increased greatly.

Extinction: After the acquisition, rats underwent 21 consecutive extinction sessions (one per day); during the phase, the number of active pokes decreased.

Reinstatement: Twenty-four hours after the last extinction session, morphine-seeking behavior was reinstated by morphine-priming injection.

#### 2.7.2. Effect of MA on Morphine Priming-Induced Reinstatement of Morphine Seeking

Rats were trained to self-administer morphine over 18 consecutive sessions. After acquisition, rats underwent 21 consecutive extinction sessions. Twenty-four hours after the last extinction session, morphine-seeking behavior was reinstated by morphine-priming injection. Rats were given MA dissolved in 0.4 mL sterile saline intravenously 45 min before the morphine-priming injection. The dose of MA is 0.4 μg/g. Control rats were injected with the same volume of saline.

#### 2.7.3. Effect of MA on the Reinstatement of Saccharin Seeking

Rats were trained to self-administer saccharin over 14 consecutive sessions. After the acquisition, rats underwent 21 consecutive extinction sessions. Forty-five minutes before the reinstatement session, rats were given MA (0.4 μg/g) dissolved in 0.4 mL sterile saline or the same volume of saline intravenously.

### 2.8. Western Blotting

Rats were decapitated, and the brain was removed. The NAc shell were dissected at −20 °C and then frozen at −80 °C. The brain tissue samples were homogenized in lysis buffer (BeyotimeP0013, Haimen, China) with protease and phosphatase inhibitor cocktails (Roche, Indianapolis, IN, USA). Protein concentrations were measured using the BCA method (Pierce, Rockford, IL, USA). A total of 10 μg of proteins were loaded in each lane and separated by 12% SDS-PAGE. The proteins were then transferred to PVDF membranes and incubated for 40 min at room temperature in a blocking solution (5% milk in Tris-buffered saline with 0.1% Tween, TBST), followed by an overnight incubation at 4 °C in blocking solutions containing primary antibodies to phosphorylated CaMKII (Thr286/287) (1:1000; Abcam, Cambridge, UK), αCaMKII (1:2000; Abcam), or βCaMKII (1:1000; Abcam). The membranes were then probed with an HRP-conjugated secondary antibody (1:3000; Zhongshan GoldenBridge Technology, Beijing, China) for 1 h at room temperature after being washed three times with TBST for five minutes each. ECL Western blotting detection reagents (Thermo Scientific, Rockford, IL, USA) allowed for the visualization of bands. ChemiDoc XRS apparatus (Bio-Rad, Hercules, CA, USA) was used to capture the signals, and Quantity One software (Bio-Rad) was used to quantify them. Following membrane stripping, an HRP-conjugated antibody to glyceraldehyde-3-phosphate dehydrogenase (GAPDH) (1:5000; Kangcheng, Shanghai, China) was used to probe the membranes as a loading control.

### 2.9. Electrophysiology in Brain Slices

Slices were prepared according to the method detailed in our earlier research [66]. Under deep anesthesia caused by sodium pentobarbital, the brain was rapidly transferred into the ice-cold cutting solution, which contained (mM) 90 sucrose, 87 NaCl, 2.5 KCl, 7 MgCl_2_, 0.5 CaCl_2_, 1.25 NaH_2_PO_4_, 25 NaHCO_3_ and 10 glucose and left for 3–5 min. Then, using a vibrating blade microtome (Leica, Heidelberger, German), 300-μm sagittal slices were cut. Slices were placed in an incubating chamber filled with artificial cerebrospinal fluid (ACSF) that contained (mM) 124 NaCl, 4.5 KCl, 1 MgCl_2_, 2 CaCl_2_, 1.25 NaH_2_PO_4_, 26 NaHCO_3_, and 10 glucose and was maintained at a pH of 7.2–7.4 bubbled with carbogen (95% O_2_/5% CO_2_). Recover was given at 33 °C for 45 min before recording. Recordings were done in a chamber, which was perfused continuously with carbonated ACSF containing 10 μM bicuculline (Tocris, Bristol, UK) to antagonize GABA_A_ receptors. Whole-cell recordings from MSNs in the shell region of NAc slices were obtained with a Heka EPC10 amplifier. Recording pipettes were filled with solution containing (mM) 122.5 Cs-gluconate, 17.5 CsCl, 2 MgCl_2_, 10 HEPES, 0.5 EGTA, 4 ATP (Sigma-Aldrich, St Louis, MO, USA) with pH 7.2–7.4 adjusted by CsOH. Excitatory postsynaptic currents (EPSCs) were evoked by stimulating prefrontal cortical inputs via constant-voltage current pulses (1 ms), which were delivered through a concentric bipolar electrode (FHC, Bowdoinham, ME, USA) at 0.067 Hz. LTD was triggered by a pairing protocol (1 Hz, 480 s, −40 mV). Neurons were voltage-clamped at −60 mV except where noted. After 7~10 min of stable baseline recording, the pairing protocol was introduced. The magnitude of LTD was estimated from EPSCs recorded during the last 10 min after LTD induction as a percentage of baseline EPSCs amplitude. We collected data by Pulse software (HEKA, Reutlingen, Germany) and transformed them by the ABF utility of Minianalysis (Synaptosoft, Decatur, GA, USA). Minianalysis was used to determine the electrophysiological properties.

### 2.10. Data Analysis

The amplitudes of the evoked EPSCs in the control group, acquisition group, extinction group, and reinstatement group were analyzed using a one-way analysis of variance (ANOVA). Dunnett’s multiple comparison test was used for the posttest. The reinstatement data were analyzed by ANOVA with treatment (MA and saline) as the between factor and poke (active and inactive) as the within factor. Other data were analyzed by independent two-tailed t-tests: the data of beam breaks, Western blot data, and comparison of EPSCs amplitude between MA group and saline controls. All the data are expressed as mean ± standard error of the mean (SEM). The level of significance was set at *p* < 0.05.

## 3. Results 

### 3.1. Alterations of the Amplitude of LTD Induced in the NAc Shell in the Different Stages of Morphine Addiction

First, we established an animal model of morphine self-administration and relapse. Rats underwent acquisition, extinction, and reinstatement of morphine-seeking behavior (see Section 2.7.1). As shown in Figure 1B, during the training course, the number of active pokes increased greatly and was maintained stably in the last three sessions (22.6 ± 3.36; 23.3 ± 2.74; 23.0 ± 2.63); during the extinction course, the number of active pokes decreased and were maintained very low at the last three extinction sessions (5.36 ± 0.800; 3.50 ± 1.09; 4.63 ± 0.844). In the reinstatement session, this number (38.8 ± 14.9) increased again. The number of inactive pokes remained stable during the last three training sessions (5.25 ± 1.41; 4.38 ± 0.706; 5.00 ± 1.83), the last three extinction sessions (3.88 ± 1.03; 3.00 ± 0.926; 4.13 ± 1.20) and the reinstatement (3.38 ± 0.981).

To determine the involvement of synaptic plasticity in the NAc shell during morphine addiction, we evaluated LTD induced in this region by the whole-cell patch–clamp. After the acquisition, extinction, and reinstatement of morphine self-administration, rats were decapitated respectively, and NAc slices were prepared. The control group was the naïve rats which were purchased at the same time. LTD was then induced in MSNs in the shell region. As shown in Figure 2, after induction, the amplitude of evoked EPSCs in the NAc shell decreased obviously in rats that acquired morphine self-administration behavior (Figure 2B) compared with the control group (Figure 2A). The amplitude of EPSCs recovered in the extinguished rats (Figure 2C). Interestingly, the amplitude of EPSCs decreased again when the morphine-seeking behavior was reinstated (Figure 2D). We analyzed the amplitude of EPSCs recorded in the last 10 min. The data (Figure 2E) revealed a significant difference between the four groups (*F*_3,36_ = 103.6, *p* < 0.001). Post analysis showed the significantly decreased amplitude of EPSCs in the acquisition group (*p* < 0.001) and the reinstatement group (*p* < 0.001), compared with the controls.

These results demonstrated the amplitude of LTD induced in the NAc shell increased after the acquisition of morphine self-administration, recovered after extinction, and increased again after reinstatement, indicating the changes in synaptic plasticity in the NAc shell play important roles in the development of morphine addiction and relapse. These results also imply that modulating the synaptic plasticity in the NAc shell may prevent the relapse to opioids.

### 3.2. Intravenous Injection of “MA” Decreases CaMKII Activity in the NAc Shell and Attenuates Morphine-Seeking Behavior

As mentioned above, some studies have demonstrated that CaMKII is involved in the mechanisms of LTD [52,53,67,68,69]. More importantly, our previous work has demonstrated that αCaMKII activity increases in the NAc shell during the reinstatement of morphine-seeking behavior, and inhibiting CaMKII activity in the NAc shell blocks the reinstatement of morphine seeking in rats [29]. Therefore, based on these results, we hypothesize that CaMKII activity in the NAc shell regulates morphine-seeking behavior by its involvement in the mechanisms of synaptic plasticity in this brain region.

To further explore the neural mechanisms of CaMKII involved in the relapse to morphine, we modified CaMKII inhibitor “myr-AIP” with poly-arginine to enhance its membrane permeability [65,70]. We obtained a new peptide named MA (see Section 2.3), which can be administrated intravenously.

Firstly, we examined whether MA had a similar blocking effect on the reinstatement of morphine-seeking behavior. After acquisition and extinction of morphine self-administration, rats were injected with MA before the reinstatement test (see Section 2.7.2). The reinstatement data (Figure 3A) was analyzed with between-within ANOVA, which revealed a main effect of MA pretreatment (*F*_1,26_ = 8.28, *p* < 0.01), and a treatment-poke interaction (*F*_1,26_ = 8.67, *p* < 0.01). Subsequent analyses showed significantly higher number of active pokes in saline controls than that in MA group (*t*_26_ = 2.99, *p* < 0.01), and no difference (*t*_26_ = 0.32, *p* = 0.76) in the number of inactive pokes. The locomotor activity was measured during the reinstatement test, and the data revealed no significant difference (*t*_22_ = 0.14, *p* = 0.89) between the saline group and MA pretreatments (Figure 3B).

Our results demonstrated that intravenous injection of MA before giving morphine-priming blocked the reinstatement of morphine-seeking behavior without influence on locomotor activity.

Then, we examined whether MA inhibited CaMKII activity during the reinstatement test. After the reinstatement test (see Section 2.7.2), rats were decapitated immediately and the brain was removed for the collection of proteins from the NAc shell. The levels of phosphorylation on Thr286/287 in CaMKII in the NAc shell were measured. Compared to the saline controls, the phosphorylated CaMKII in the NAc shell of the MA group decreased (Figure 4B,D), whereas the levels of total αCaMKII and βCaMKII did not change (Figure 4A,C). Because phosphorylation of CaMKII on Thr286/287 means the activity of the enzyme increases, our data revealed intravenous injection of MA inhibited CaMKII activity in the NAc shell during the reinstatement test.

In summary, the modified peptide “MA” blocked the reinstatement of morphine-seeking through its inhibitory effect on the CaMKII activity in the NAc shell. Because MA can be administrated intravenously, it has potential clinical value in preventing relapse to opioids.

### 3.3. LTD Expression in MSNs in the NAc Shell Was Absent in MA-Pretreated Rats

To determine whether changes in synaptic plasticity in the NAc shell involved in the inhibitory effect of MA on the reinstatement of morphine seeking, we evaluated LTD induced in the NAc shell in the rats pretreated with MA.

After the reinstatement test (see Section 2.7.2), rats were decapitated immediately and NAc slices were prepared. LTD was then induced in MSNs in the shell region. In the saline controls, LTD was induced robustly (Figure 5A); however, the expression of LTD was absent in the MA group (Figure 5B). We analyzed the amplitude of EPSCs recorded in the last 10 min. The data revealed a significant difference (*t*_138_ = 10.9, *p* < 0.001) between the MA group and the saline controls (Figure 5C).

Consistent with our hypothesis, these results suggest that increased CaMKII activity in the NAc shell promotes morphine-seeking behavior through its involvement in regulating synaptic plasticity in this brain region. Decreasing CaMKII activity in the NAc shell by MA pretreatment impairs the synaptic plasticity, reflected in the absence of LTD in this region.

### 3.4. MA Had No Significant Influence on the Reinstatement of Saccharin-Seeking Behavior

To determine whether MA attenuated morphine-seeking behavior specifically without affecting the natural reward process, we examined its effect on the saccharin-seeking behavior (see Section 2.7.3). As shown in Figure 6, the data revealed there were no significant differences (*F*_1,15_ = 3.21, *p* = 0.09) in the number of noses pokes during the reinstatement. This result demonstrated MA pretreatment did not influence the saccharin-seeking behavior, which means MA can be a potential medication for preventing relapse to opioids without side effects on the natural reward process.

## 4. Discussion 

### 4.1. Opioid Addiction and Synaptic Plasticity in the NAc Shell

In this study, we found the alterations in the synaptic plasticity (also called metaplasticity [71]) in the NAc shell were closely associated with the development of opioid addiction behavior, reflected in the changes of LTD expression in the NAc shell during the different stages of morphine self-administration. Moreover, pretreatment of CaMKII inhibitor MA suppressed the expression of LTD in the NAc shell and blocked the reinstatement of morphine-seeking behavior induced by morphine priming. These results demonstrated that repeated drug exposure potentiated the synaptic plasticity in the NAc shell, which may be an important part of the mechanisms underlying the durable neurotransmission changes and high risk of relapse. During extinction training, synaptic plasticity in the shell was recovered in the absence of morphine but easily to be altered. During the reinstatement, only a small dose of morphine priming caused the rapid change of synaptic plasticity in the shell, contributing to the reinstatement of morphine-seeking behavior. Therefore, alteration of synaptic plasticity in the NAc shell can be considered as a switch, potentiation of synaptic plasticity represents “on” and impairment and even loss represent “off”. In the “on” state, drug-seeking behavior is prompted, whereas in the “off” state, it is suppressed. Results from other related studies also support our inference. For example, Li et al. found synaptic plasticity in the NAc shell was potentiated following morphine-induced conditioned place preference (CPP) expression in rats, demonstrated by the facilitated LTP induced in the pathway from the hippocampus to the NAc shell [72].

Moreover, alteration of synaptic plasticity in the NAc shell can regulate drug-seeking behavior, offering a potential target mechanism for the treatment of drug addiction and prevention of relapse. Interventions that potentiate or weaken synaptic plasticity in the NAc shell and even other brain regions can regulate drug-seeking behavior. In fact, results from some studies support this opinion. For example, Brebner et al. found systemic or intra-NAc infusion of the membrane-permeable GluR2 peptide, which blocks NAc LTD, prevented the expression of amphetamine-induced behavioral sensitization in the rat [28]. Usage of “MA” in this study blocked expression of LTD in the NAc shell and attenuated morphine-induced reinstatement of drug-seeking behavior.

### 4.2. CaMKII Activity and Expression of LTD

Our previous studies have demonstrated CaMKII activity in the NAc shell increased during the relapse to morphine [30], and inhibition of CaMKII activity blocked the reinstatement of morphine-seeking behavior [29]. In this paper, we found that increased CaMKII activity in the NAc shell was involved in the alteration of synaptic plasticity during the morphine-priming induced relapse, demonstrated by the absence of LTD in the NAc shell of rats pretreated with “MA”. The modified peptide “MA”, which decreased CaMKII activity and attenuated the relapse to morphine-seeking behavior, blocked the expression of LTD in the NAc shell. Based on these results, we infer that CaMKII-mediated potentiation of synaptic plasticity in the NAc shell plays an important role in the relapse to opioids induced by a small dose of the drug. Exposure to morphine priming induced increase of CaMKII activity in the NAc shell, then facilitated the potentiation of synaptic plasticity (switch “on”), and finally resulted in the reinstatement of morphine- seeking behavior.

It is well known that CaMKII activity is involved in the mechanisms of synaptic plasticity and memory processing [56]. In the last decade, research has revealed the involvement of CaMKII activity in LTD and associated molecular mechanisms. Mockett et al. found that Group I mGluR-dependent protein synthesis and associated LTD in rat hippocampus required the activation of CaMKII [73]. Coultrap et al. found that NMDAR-dependent LTD required “autonomous” CaMKII activity-mediated phosphorylation of AMPA receptor subunit GluR1 at Ser567 [68]. Goodell et al. found that activation of DAPK1 suppressed the binding of CaMKII to GluN2B and its synaptic accumulation, which is crucial for LTD [74]. More interestingly, Woolfrey et al. found a novel regulation mechanism in which CaMKII regulates the depalmitoylation and synaptic removal of the scaffold protein AKAP79/150 indirectly to mediate structural long-term depression correlated with dendritic spine shrinkage in cultured hippocampal neurons [53]. These studies indicate that CaMKII can mediate both LTP and LTD through differential trafficking and substrate selection [56]. In our study, the mechanisms of CaMKII in the expression of LTD are still unclear and need further studies to illuminate them.

### 4.3. The Specific Inhibitory Effect of MA on Morphine-Seeking Behavior

Our previous study found protein levels of αCaMKII phosphorylated on Thr286 and βCaMKII phosphorylated on Thr287 in the NAc core decreased but did not change in the NAc shell during the reinstatement of saccharin-seeking behavior induced by saccharin deliveries [30]. Now that CaMKII activity did not increase in the NAc shell, and even decreased in the NAc core, it is not surprising that CaMKII inhibitor “MA” did not attenuate the saccharin-seeking behavior. These results also reveal the effects of drugs of abuse on reward-related circuits are completely different from natural rewards, not only in the change of molecules but in the involved subregions.

The specific effect on drug-seeking behavior makes MA a potential medication for preventing relapse to opioids. However, because CaMKII is multifunctional and is expressed ubiquitously in the brain and other tissue [56], inhibition of CaMKII through intravenous injection of MA may cause some side effects.

### 4.4. CaMKII Activity and MA

Activated CaMKII subunit can be phosphorylated on Thr286 (this numbering refers to the α-isoform and is Thr287 in the β-isoform) by its neighboring active subunit, making it has Ca^2+^-independent “autonomous” activity [56]. When lots of kinases are phosphorylated in the site, the rate of autophosphorylation will exceed the rate of dephosphorylation, leading to the long-term persistent activation of CaMKII [55]. As Myr-AIP or AIP inhibits the enzyme activity by binding to the substrate-binding site of the CaMKII subunit for autophosphorylation [75], it reduces the protein level of CaMKII phosphorylated on Thr286 (Thr287) without effect on the expression of total αCaMKII and βCaMKII. MA is obtained by modifying myr-AIP with poly-arginine, so it has a similar effect (Figure 4) to myr-AIP.

## 5. Conclusions

Our research data suggest that alterations of synaptic plasticity in the NAc shell play an important role in the development of morphine addiction and the relapse to morphine-seeking behavior in rats, demonstrated by the changes in LTD expression in different stages of addiction. Increased CaMKII activity in the NAc shell involved potentiation of synaptic plasticity and the reinstatement of morphine-seeking behavior. Intravenous injection of MA decreased CaMKII activity in the NAc shell, therefore inhibited the potentiation of synaptic plasticity in this brain region, demonstrated by the absence of LTD expression, and finally blocked the relapse to morphine-seeking behavior induced by morphine priming. MA has no significant effect on the locomotor activity and natural reward process, implying its value in the prevention of relapse to opioids.

## Figures and Tables

**Figure 1 brainsci-12-00985-f001:**
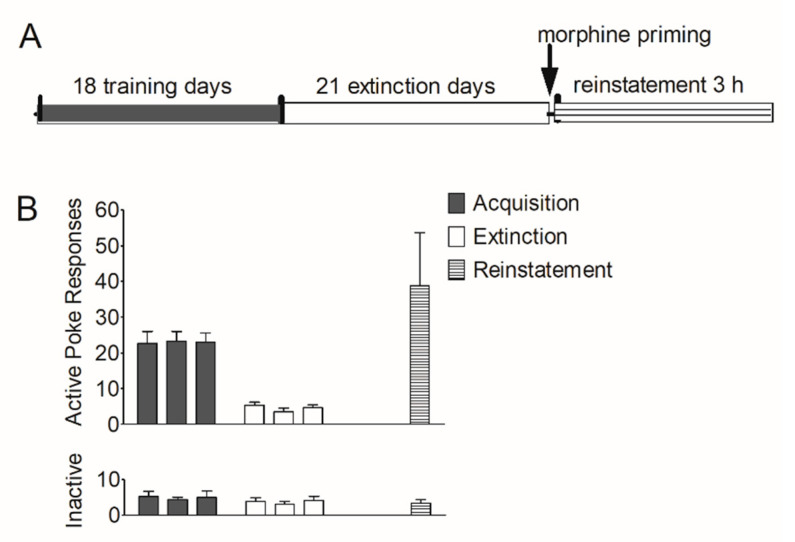
Acquisition, extinction, and reinstatement of morphine-seeking behavior in rats. (**A**) Timeline of acquisition, extinction, and morphine priming-induced reinstatement in rats. (**B**) The number of active pokes and inactive pokes in the last 3 training sessions, the last 3 extinction sessions, and the reinstatement session (*n* = 8). Data are expressed as mean ± SEM.

**Figure 2 brainsci-12-00985-f002:**
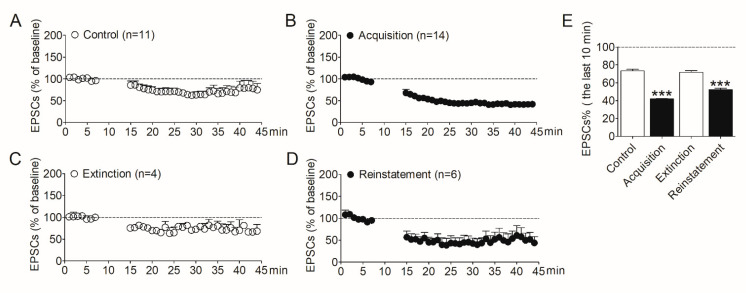
Alterations of LTD expression in the NAc shell were correlative to the addiction behavior in the animal model of morphine self-administration. (**A**) LTD was induced in the NAc shell in the control rats (*n* = 11). (**B**) LTD induced in the NAc shell in the rats acquired morphine self-administration (*n* = 14). (**C**) LTD induced in the NAc shell in the extinguished rats (*n* = 4). (**D**) LTD induced in the NAc shell in the rats reinstated by morphine priming (*n* = 6). (**E**) Average EPSCs (% of baseline) recorded in the last 10 min after LTD induction in the four groups. Asterisks indicate significant decreases of EPSCs in the acquisition and reinstatement groups compared to the controls (*** *p* < 0.001). Data are expressed as mean ± SEM.

**Figure 3 brainsci-12-00985-f003:**
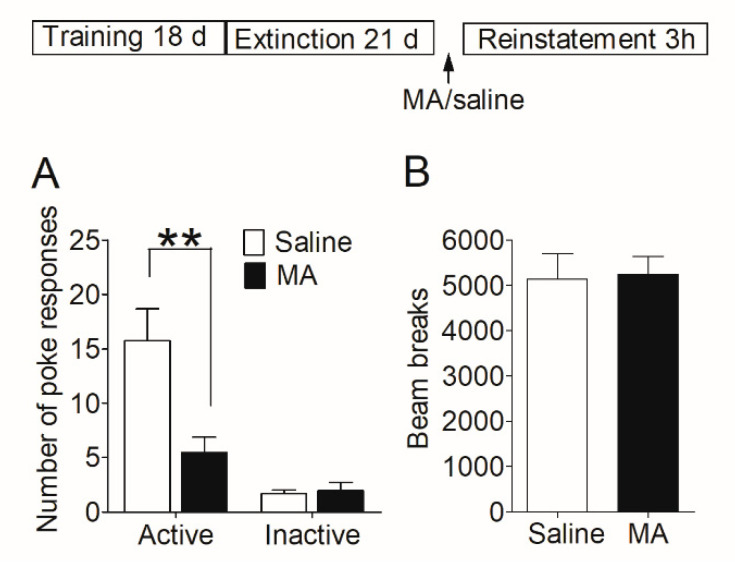
Intravenous injection of MA before the reinstatement test blocked the reinstatement of morphine seeking induced by morphine-priming injection and had no significant effect on locomotor activity. (**A**) Number of nose-pokes in saline group (*n* = 15) and MA group (*n* = 13) in the reinstatement session. Asterisks indicate a significant difference (*** p* < 0.01) in the number of active pokes between MA group and saline controls. (**B**) The level of locomotor activity in the reinstatement session. There was no significant difference between the two groups. Data are expressed as mean ± SEM.

**Figure 4 brainsci-12-00985-f004:**
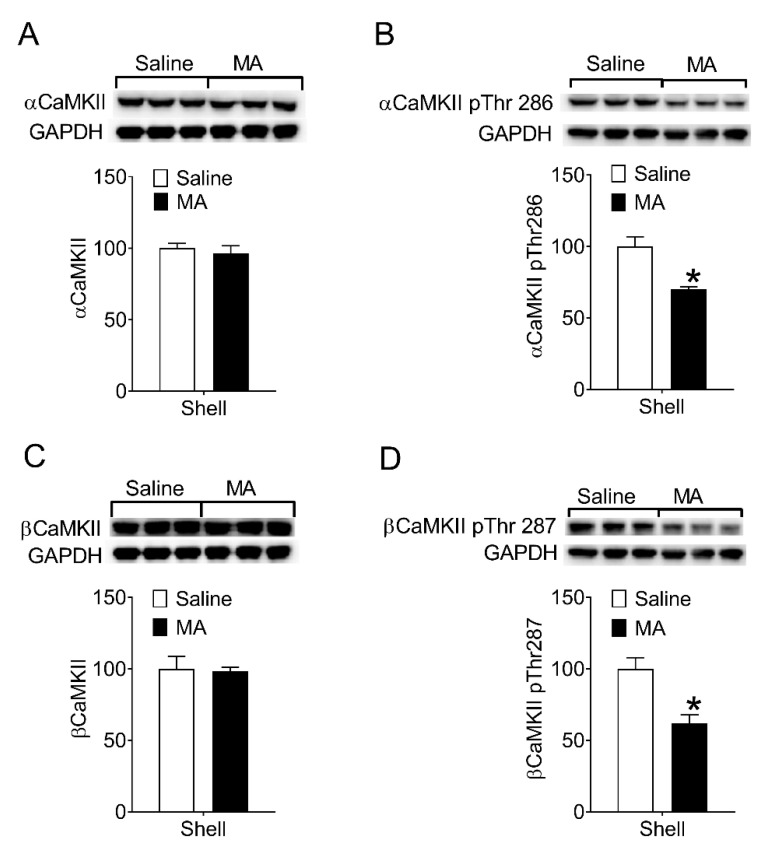
Intravenous injection of MA decreased phosphorylation of CaMKII on Thr286/287 in the NAc shell during the reinstatement. (**A**,**C**) No significant differences in the levels of total α (*t*_4_ = 0.56, *p* = 0.61) and βCaMKII (*t*_4_ = 0.16, *p* = 0.88) between saline controls (*n* = 3) and MA group (*n* = 3). (**B**) The levels of αCaMKII phosphorylated on Thr286 in the NAc shell were lower in the MA group than that in the saline controls during the reinstatement (*t*_4_ = 4.23, *p* < 0.02). (**D**) The levels of βCaMKII phosphorylated on Thr287 in the NAc shell were lower in the MA group than that in the saline controls during the reinstatement (*t*_4_ = 3.84, *p* < 0.02). Asterisks indicate a significant difference (* *p* < 0.05) between saline controls and MA group. Data are expressed as mean ± SEM.

**Figure 5 brainsci-12-00985-f005:**
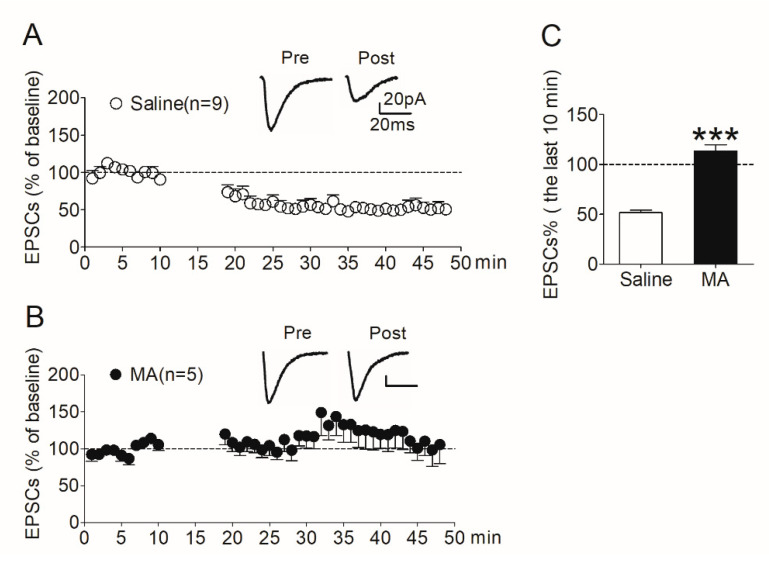
LTD induced in the NAc shell was absent in rats pretreated with MA. (**A**) LTD was induced robustly (51.8 ± 2.34% of baseline, *n* = 9) in saline controls. (**B**) LTD was absent (113 ± 6.31% of baseline, *n* = 5) in MA-pretreated rats. (**C**) Average EPSCs (% of baseline) recorded in the last 10 min after induction in the two groups. Asterisks indicate a significant difference (*** *p* < 0.001) between saline controls and MA group. Data are expressed as mean ± SEM.

**Figure 6 brainsci-12-00985-f006:**
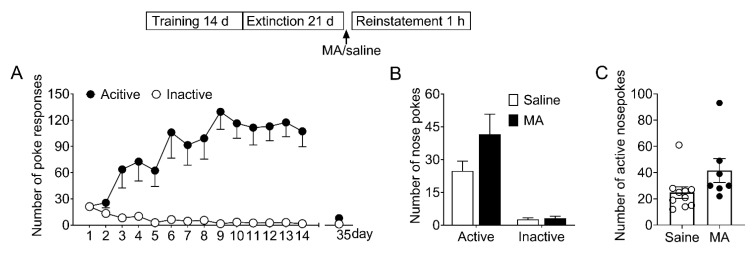
Intravenous injection of MA before the reinstatement test had no significant effect on the reinstatement of saccharin-seeking behavior induced by noncontingent saccharin delivery. (**A**) Number of nose-pokes in the 14 training sessions (days 1–14) and the last extinction session (day 35). (**B**) Number of nose-pokes in saline group (*n* = 10) and MA group (*n* = 7) in the reinstatement session. There was no significant difference in the number of nose-pokes between the two groups. Data are expressed as mean ± SEM. (**C**) Number of active nose-pokes of each individual in the reinstatement session. MA: *n* = 7; Saine: *n* = 10.

## Data Availability

The data presented in this study are available on request from the corresponding author.

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
