# Peer review of "A Novel CaMKII Inhibitory Peptide Blocks Relapse to Morphine Seeking by Influencing Synaptic Plasticity in the Nucleus Accumbens Shell"

_brainsci, 2022, doi:10.3390/brainsci12080985_

Round 1
Reviewer 1 Report
Craving and compulsive behavior for drugs are often a culmination of long-term changes in synaptic transmission in brain reward circuits. A possibility of relapse cannot be completely ruled out even if detoxification is achieved. Mechanisms of relapse could be diverse and addressing these could prove to be vital in developing effective therapeutic strategies. In the current manuscript, Liu et al illustrate the synaptic mechanisms of relapse to morphine seeking behavior in an animal model of morphine self-administration. The authors developed a novel peptide obtained by modifying a CamKII inhibitor ‘myr-AIP’ and showed that an intravenous injection of this novel peptide, MA decreased CamKII activity in the NAc shell and blocked reinstatement of the morphine seeking behavior. LTD was shown to be completely absent in the NAc shell of rats which were pre-treated with MA while it was intact in control rats in which the morphine seeking behavior was reinstated. The manuscript is well written and timely. The authors have performed good, directed experiments to support their hypothesis. I recommend minor revision for this manuscript, primarily because of the following reasons:
1. In Figure 4B, can the authors provide a better representative example? In this one, it doesn't necessarily reflect the data where the authors are claiming that MA treatment reduces αCamKIIpThr286 in the NAc shell.
2. In Figure 6B, can the authors show individual data points for the MA treatment in the active group? I am not sure why this is not significantly different than saline treated rats.
Author Response
We appreciate you for the valuable comments and suggestions, which are all very helpful for improving our manuscript. We have studied the comments carefully and have made corrections. We hope the corrections can meet with approval.
Point 1: In Figure 4 B, can the authors provide a better representative example? In this one, it doesn't necessarily reflect the data where the authors are claiming that MA treatment reduces αCamKII pThr286 in the NAc shell.
Response 1: Thanks for your suggestion. We have changed Figure 4 B in the revised manuscript.
Point 2: In Figure 6 B, can the authors show individual data points for the MA treatment in the active group? I am not sure why this is not significantly different than saline treated rats.
Response 2: We appreciate this comment and agree with the reviewer. We have shown the individual data points in the revised Figure 6.

Reviewer 2 Report
In their manuscript, Liu and colleagues report that a novel peptide obtained by modifying Ca2+/calmodulin-dependent protein kinase II (CaMKII) inhibitor ‘myr-AIP’ blocked relapse to morphine seeking by influencing synaptic plasticity in the nucleus accumbens shell. The results are interesting and may interest the journal’s readership. However, some points should be addressed before publication:
Major points
· The Authors need to rewrite better the manuscript. There are several grammatical errors that obstacle the reading as well as the understanding of the manuscript.
· How did the Authors choice to inject MA at the dose of 0.4μg/g? The authors need to explain this or add some references supporting the choice of this dose.
· “MA had no influence on the reinstatement of saccharin seeking behavior”. In this respect there is an important trend in the figure 6B suggesting that MA could increase the saccharin seeking behavior. Moreover, the error bar of MA treated group is high. Is there any outlier? This has important implications.
Minor points
· The title must be more specific reporting that this novel peptide is a CaMKII inhibitor.
· The authors should discuss more the role of NAc in reward related behaviors as well as in drug addiction. I suggest to add the following reference (PMID: 30648615) that can remark the complexity of the NAc-dependent mechanisms controlling reward-related behaviors, and also the following references (PMID: 27363441; 19710631) for drug addiction.
· The statistical values of ANOVA (F values and p values) need to be reported besides all the results.
· The potential therapeutic implications reported by the Authors are interesting. However it is also important to report that this kind of inhibition might induce off targets effects and thus severe side effects related to the physiological role of CaMKII.
Author Response
We appreciate you for the valuable comments and suggestions, which are all very helpful for improving our manuscript. We have studied the comments carefully and have made corrections. We hope the corrections can meet with approval.
Major points
Point 1: The Authors need to rewrite better the manuscript. There are several grammatical errors that obstacle the reading as well as the understanding of the manuscript.
Response 1: Thanks for your suggestion. A native English speaker has edited the revised version.
Point 2: How did the Authors choice to inject MA at the dose of 0.4μg/g? The authors need to explain this or add some references supporting the choice of this dose.
Response 2: We thank the reviewer for this comment. We chose this dose based on our previous published study(Liu, Zhang et al. 2012), in which the dose of myr-AIP injected into the NAc shell was 24 nmol for each rat (about 350g). Therefore, we injected MA intravenously at 0.4μg/g to examine its effect. At this dose, rats weighing 350g received about 48 nmol MA. We found this dose was effective not only for inhibiting CaMKII activity in the brain but also for attenuating the relapse to morphine. Based on the trial, we used the dose in all MA-pretreated rats.
Point 3: “MA had no influence on the reinstatement of saccharin seeking behavior”. In this respect there is an important trend in the figure 6B suggesting that MA could increase the saccharin seeking behavior. Moreover, the error bar of MA treated group is high. Is there any outlier? This has important implications.
Response 3: Thank you very much for these helpful comments. We have shown the individual data points in the revised Figure 6 and discussed it in Discussion section 4.3.
Minor points
Point 1: The title must be more specific reporting that this novel peptide is a CaMKII inhibitor.
Response 1: Thanks for the suggestion. We have changed the title in the revised manuscript.
Point 2: The authors should discuss more the role of NAc in reward related behaviors as well as in drug addiction. I suggest to add the following reference (PMID: 30648615) that can remark the complexity of the NAc-dependent mechanisms controlling reward-related behaviors, and also the following references (PMID: 27363441; 19710631) for drug addiction.
Response 2: We thank the reviewer for this comment. We have revised the introduction section and cited the suggested references.
Point 3: The statistical values of ANOVA (F values and p values) need to be reported besides all the results.
Response 3: We appreciate this comment. In the revised manuscript, we showed all F values or t values, and p values in the main text or in the figure legend.
Point 4: The potential therapeutic implications reported by the Authors are interesting. However, it is also important to report that this kind of inhibition might induce off targets effects and thus severe side effects related to the physiological role of CaMKII.
Response 4: We thank the reviewer for this helpful comment. In the revised manuscript, we mentioned it in Discussion section 4.3.
Reference
Liu, Z., J. J. Zhang, X. D. Liu and L. C. Yu (2012). "Inhibition of CaMKII activity in the nucleus accumbens shell blocks the reinstatement of morphine-seeking behavior in rats." Neurosci Lett 518(2): 167-171.

Reviewer 3 Report
Comments to Authors
The present study, “A novel peptide blocks relapse to morphine seeking by influencing synaptic plasticity in the nucleus accumbes shell”, examined how morphine induced reinstetment and LTD and how a new peptide, MA, reduced reinstatement behavior and LTD. Moreover, the study labled alphaCaMKII, betaCaMKII, alphaCaMKII pThr 286, and betaCaMKII pThr 287 in the nucleus accymbens shell using Western blot after morphine-induced reinstatement and LTD. The study showed that morphine induced resintatement and LTD; MA reduced morphine-induced reinstatement and LTD. MA reduced alphaCaMKII pThr 286 and betaCaMKII pThr 287 in the nucleus accumbens shell; however, MA did not affect alphaCaMKII and betaCaMKII in the nucleus accumbes shell. In particular, the results showed that MA did not affect saccharin self-administration-induced reinstatement behavior. The present data are inetersting and performed a lot of work. However, some points should be concerned as follows.
Major points:
1. In Figure 6, why does saccharin SA show a nonsignificant number of nose pokes between the saline and MA groups? Why is this effect different from those of morphine SA? It should be clarified in the Discussion section.
2. Why can MA reduce morphine-induced LTD and reinstatement effects?
3. How is morphine-induced LTD related to relapse behavior in the reinstatement testing phase?
4. Why does the study label alphaCaMKII pThr 286 and betaCaMKII pThr 287 in the nucleus accumbens shell? What is its rationale?
5. Explain why alphaCaMKII and betaCaMKII did not show significant differences and why alphaCaMKII pThr 286 and betaCaMKII pThr 287 show significant differences between the saline and MA groups in the Discussion section.
Minor points:
1. The results in Figure 4 should show the F and p values of the statistical analysis in Lines 369-377.
2. All figures should indicate their n numbers for each group.
3. In Figure 6B, the results should show the values of p and t. Why were there no significant differences in the number of noses pokes between the Salien and MA groups? It should show its statistical values, such as t and p values.
The current status of the present study is not good enough for publication. The study should be considered for major revision.
Author Response
We appreciate you for the valuable comments and suggestions, which are all very helpful for improving our manuscript. We have studied the comments carefully and have made corrections. We hope the corrections can meet with approval.
Major points:
Point 1:In Figure 6, why does saccharin SA show a nonsignificant number of nose pokes between the saline and MA groups? Why is this effect different from those of morphine SA? It should be clarified in the discussion section.
Response 1: We thank the reviewer for this comment. In the revised manuscript, we showed the individual data points in Figure 6 and explained them in the discussion section.
Point 2:Why can MA reduce morphine-induced LTD and reinstatement effects?
Response 2: We appreciate this comment.
(1) Our previous published study(Liu, Liu et al. 2012) has demonstrated that CaMKII activity in the NAc shell increased during the reinstatement, so MA blocks reinstatement by inhibiting the increase of CaMKII activity in the NAc shell.
(2) We discussed the involvement of CaMKII activity in the LTD and associated molecular mechanisms revealed by the research in the last decade in discussion section 4.2. MA can inhibit CaMKII activity and then influences the expression of LTD. However, the cellular and molecular mechanisms of CaMKII involved in the expression of LTD induced in the NAc shell during the reinstatement are still unclear and need further studies. We hope we can clarify them in future research.
Point 3: How is morphine-induced LTD related to relapse behavior in the reinstatement testing phase?
Response 3: We thank the reviewer for this comment. LTD is an important mechanism for synaptic plasticity and neuroadaptation induced by internal and external stimulants. Drug-evoked changes in synaptic plasticity in the reward-related circuits are important mechanisms of drug addiction and relapse. During the reinstatement phase, the amplitude of LTD induced in the NAc shell increased, indicating morphine priming caused the potentiation of synaptic plasticity in the NAc shell, which may represent an “on” state facilitating drug-seeking behavior. We discussed more details in Discussion section 4.1. The related molecular mechanism and involved pathway need further studies to elucidate.
Point 4: Why does the study label alphaCaMKII pThr 286 and betaCaMKII pThr 287 in the nucleus accumbens shell? What is its rationale?
Response 4: We appreciate this comment. (1) Our previous published study(Liu, Liu et al. 2012) has demonstrated that alphaCaMKII pThr 286 in the nucleus accumbens shell increases during priming-induced reinstatement of morphine-seeking, so in this study, we further explored the topic. (2) Phosphorylated CaMKII on Thr 286 has Ca2+-independent “autonomous” activity. MA reduced CaMKII activity by inhibiting autophosphorylation of CaMKII and has no selective effect for alpha and beta isoforms, so we measured the level of both alphaCaMKII pThr 286 and betaCaMKII pThr 287 in the NAc shell to examine the inhibitory effect of MA on the enzyme activity.
Point 5: Explain why alphaCaMKII and betaCaMKII did not show significant differences and why alphaCaMKII pThr 286 and betaCaMKII pThr 287 show significant differences between the saline and MA groups in the Discussion section.
Response 5: We thank the reviewer for this comment. We have discussed it in the revised Discussion section (see the main text 4.4).
Minor points:
Point 1: The results in Figure 4 should show the F and p values of the statistical analysis in Lines 369-377.
Response 1: We appreciate this comment. In the revised manuscript, we showed F values or t values, and p values in the main text or the figure legend.
Point 2: All figures should indicate their n numbers for each group.
Response 2: We thank the reviewer for this comment. In the revised manuscript, we showed the n numbers for each group in the figure legend.
Point 3: In Figure 6 B, the results should show the values of p and t. Why were there no significant differences in the number of noses pokes between the Saline and MA groups? It should show its statistical values, such as t and p values.
Response 3: We appreciate this comment. In the revised manuscript, we showed the individual data points in Figure 6 and their statistical values in the main text.
Reference
Liu, Z., X. D. Liu, J. J. Zhang and L. C. Yu (2012). "Increases in alphaCaMKII phosphorylated on Thr286 in the nucleus accumbens shell but not the core during priming-induced reinstatement of morphine-seeking in rats." Neurosci Lett 526(1): 39-44.

Round 2
Reviewer 2 Report
The Authors have successfully addressed all the issues I raised.